# Thermodynamically-Consistent Modeling of Ferromagnetic Hysteresis

**DOI:** 10.3390/ma16072882

**Published:** 2023-04-04

**Authors:** Claudio Giorgi, Angelo Morro

**Affiliations:** 1Dipartimento di Ingegneria Civile Ambiente Territorio Architettura e Matematica, Università di Brescia, Via Valotti 9, 25133 Brescia, Italy; 2Dipartimento di Informatica, Bioingegneria, Robotica e Ingegneria dei Sistemi, Università di Genova, Via All’Opera Pia 13, 16145 Genova, Italy; angelo.morro@unige.it

**Keywords:** magnetization, ferromagnetic hysteresis, magnetic susceptibility, thermodynamic consistency, rate equations, 74A20, 74D10, 74F15, 74A15, 74N30, 78A25, 80A17

## Abstract

Models of ferromagnetic hysteresis are established by following a thermodynamic approach. The class of constitutive properties is required to obey the second law, expressed by the Clausius–Duhem inequality, and the Euclidean invariance. While the second law states that the entropy production is non-negative for every admissible thermodynamic process, here the entropy production is viewed as a non-negative constitutive function. In a three-dimensional setting, the magnetic field and the magnetization are represented by invariant vectors. Next, hysteretic properties are shown to require that the entropy production is needed in an appropriate form merely to account for different behavior in the loading and the unloading portions of the loops. In the special case of a one-dimensional setting, a detailed model is determined for the magnetization function, in terms of a given susceptibility function. Starting from different initial magnetized states, hysteresis cycles are obtained by solving a nonlinear ordinary differential system. Cyclic processes with large and small amplitudes are established for materials such as soft iron.

## 1. Introduction

Hysteresis is a phenomenon relevant to various areas of science and means that the non-linear relation between two physical quantities, say input and output, changes depending on the increasing or decreasing phase of the input. In particular, ferro-magnetic hysteresis phenomena, along with the variation in magnetic susceptibility, affect the positional accuracy in magnetic resonance imaging systems [1] and occur during a typical charge-and-discharge process of a high-temperature superconducting magnet for NMR applications [2]. Hence, much effort has been devoted to the reduction and correction of magnetic hysteresis in magnetic-resonance imaging devices.

The first detailed model of hysteresis traces back to Duhem [3]. If *u* is a piecewise monotone input, then the output *x* is given by
(1)$$\dot{x}\left(t\right)=\{\begin{array}{cc} \phi_{l}\left(x\left(t\right),u\left(t\right)\right)\dot{u}\left(t\right), & \mathrm{for}\;\dot{u}\left(t\right)\leq 0,\\ \phi_{r}\left(x\left(t\right),u\left(t\right)\right)\dot{u}\left(t\right), & \mathrm{for}\;\dot{u}\left(t\right)\geq 0, \end{array}$$
where a superposed dot denotes the time derivative. Duhem-like models have been developed and investigated in several contexts, such as circuit theory [4,5] and ferromagnetic materials [6,7]. Next Preisach [8] modeled hysteresis by introducing two thresholds characteristic of the material [9,10]. Lately, further models have been developed by means of the Langevin function [11,12] and potential functions [13]. A generalization of the Preisach model was investigated in [14,15] through hysteresis operators, and a connection with thermodynamics was developed through hysteretic (clockwise and counterclockwise) potentials and a dissipation operator.

Duhem-like rate equations seem to be the most convenient schemes for describing any type of hysteresis. Moreover, to our mind, once the balancing (dynamic) laws of a continuum are established, the second law of thermodynamics has to be the key point to characterizing admissible constitutive properties. Following that, in this paper, we develop a thermodynamic approach to ferromagnetic hysteresis by requiring the consistency of the constitutive functions with the second law expressed by the Clausius–Duhem inequality. Indeed, while the second law states that the entropy production, say γ, is non-negative for every admissible thermodynamic process, we follow the assumption that γ itself has to be considered as given by a constitutive function, the entropy η and the other constitutive functions. This view in essence traces back to Green and Naghdi [16], though thereafter no significant application has been developed in the literature. Lately, we have made recourse to this scheme in connection with hysteresis in plasticity [17] and ferroelectrics [18].

The purpose of this paper is to establish a model of hysteresis for ferromagnetic materials. First, general thermodynamic relations are expressed in a three-dimensional setting. The (ferromagnetic) body is allowed to be deformable, and hence, balance equations and constitutive assumptions involve mechanical and electromagnetic properties. Since hysteresis is determined by a non-linear relation between the rate of magnetization and the magnetic field, it is non-trivial to comply with the objectivity principle, whereby the constitutive equations are required to be invariant relative to Euclidean transformations. It follows that both objectivity and the balance of angular momentum hold if the magnetization and magnetic field are expressed by Lagrangian fields.

Next, with the restriction to collinear fields, we establish explicit models of hysteresis suitable for describing soft iron materials. As a thermodynamic restriction, it follows that the hysteresis curve is run in the counterclockwise sense. Examples are given of cycles with different properties of the asymptotic regime (saturation).

### Notation

We consider a body occupying a time-dependent region $\Omega \subset E^{3}$. The motion is described by means of the function χ(X,t), providing the position vector x∈Ω=χ(R,t). The symbols ∇ and ∇R denote the gradient operator with respect to x∈Ω and X∈R. The function χ is assumed to be differentiable; hence, we can define the deformation gradient as F=∇Rχ, or in suffix notation, $F_{iK}=\partial_{X_{K}}\chi_{i}$. The invertibility of X→x=χ(X,t) is guaranteed by letting J:=detF>0. For any tensor A, we define |A| as ${\left(A\cdot A\right)}^{1/2}$. Throughout (x,t)∈Ω×R. We let v(x,t) be the velocity field. For any function f(x,t), we let $\dot{f}$ be the total time derivative; $\dot{f}=\partial_{t}f+\left(v\cdot \nabla \right)f$. A prime denotes the derivative of a function with respect to the argument.

## 2. Balance Equations

We consider a ferromagnetic, deformable body where electric conduction and electric polarization are negligible. Let ρ(x,t) be the mass density. The balance of mass leads to the local continuity equation
$$\dot{\rho }+\rho \nabla \cdot v=0.$$
Let T be the mechanical Cauchy stress tensor and b be the mechanical body force. The equation of motion can be written in the form
$$\rho \dot{v}=\nabla \cdot T+\rho b+f_{M},$$
where $f_{M}$ is the force per unit volume of magnetic character. In stationary conditions or in the approximation of a negligible electric field, we have
$$f_{M}=\mu_{0}\left(M\cdot \nabla \right)H$$
where H is the magnetic field, M the magnetization (per unit volume), and $\mu_{0}$ the permeability of free space. The balances of angular momentum and energy are taken in the form
(2)$$\mathrm{skw}(T+\mu_{0}H⊗M)=0,$$
(3)$$\rho \dot{\varepsilon }=\mu_{0}\rho H\cdot \dot{m}+T\cdot L-\nabla \cdot q+\rho r,$$
where ε is the internal energy (per unit mass), m=M/ρ, L is the velocity gradient, $L_{ij}=\partial_{x_{j}}v_{i}$, q is the heat-flux vector, and *r* is the energy supply (per unit mass).

Let η be the entropy density and θ the absolute temperature. As the second law of thermodynamics, we take the following statement: the inequality
(4)$$\rho \dot{\eta }+\nabla \cdot \frac{q}{\theta }-\frac{\rho r}{\theta }=\rho \gamma \geq 0$$
holds for any process compatible with the balance equations. The non-negative scalar γ, namely, the (rate of) entropy production per unit mass, is assumed to be given by a constitutive function. Hence, the thermodynamic process consists of η,q,r,γ, and the other functions occurring in the balance equations.

In terms of the Helmholtz free energy
ψ=ε−θη
the entropy (or Clausius–Duhem) inequality (Equation 4) can be written as
(5)$$-\rho \left(\dot{\psi }+\eta \dot{\theta }\right)+\mu_{0}\rho H\cdot \dot{m}+T\cdot L-\frac{1}{\theta }q\cdot \nabla \theta =\theta \rho \gamma \geq 0.$$

To simplify the description of the material properties, it is understood that H and M are the fields measured in the reference locally at rest with the body.

## 3. Euclidean Invariance and Power Representation

The internal energy ε, the entropy η, and the free energy ψ are invariant under a change of frame. Hence, they can depend only on invariant quantities. A change in frame $F\to F^{*}$ given by a Euclidean transformation, such that $x\mapsto x^{*}$, is expressed by
(6)$$x^{*}=c+Qx,\;Q^{T}Q=1.$$

Under the transformation (Equation 6), the deformation gradient F and the magnetic field H change as vectors:$$F^{*}=QF,\;H^{*}=QH,$$
and hence they are not invariant. Yet invariant scalars, vectors, and tensors occur in connection with F and H.

We first look at invariants of mechanical character. The right Cauchy–Green tensor C and the Green–St. Venant strain tensor E, defined as
$$C=F^{T}F,\;E=\frac{1}{2}\left(C-1\right),$$
are invariant in that
$$C^{*}=F^{*}{}^{T}F^{*}=F^{T}Q^{T}QF=F^{T}F=C,$$
and the like for E. Consequently, the scalar
F·F=trC=2trE+3
is invariant too. Since
$$L^{*}=QLQ^{T}+\dot{Q}Q^{T},$$
the power T·L is apparently non-invariant. Decompose L in the classical form
L=D+W,
where D is the stretching tensor and W is the spin; we have
$$D^{*}=QDQ^{T},\;W^{*}=QWQ^{T}+\dot{Q}Q^{T}.$$

Let
$$T_{RR}=JF^{-1}TF^{-T}$$
be the second Piola stress. We observe that since $\dot{E}=F^{T}DF$, so
$$T\cdot D=J^{-1}\left(FT_{RR}F^{T}\right)\cdot D=J^{-1}T_{RR}\cdot \left(F^{T}DF\right)=J^{-1}T_{RR}\cdot \dot{E}.$$

Hence, we have
(7)$$T\cdot L=J^{-1}T_{RR}\cdot \dot{E}+T\cdot W.$$

The referential heat flux and temperature gradient
$$q_{R}=JF^{-1}q,\;\nabla \;R\;\theta =F^{T}\nabla \theta$$
are invariant, and so is the power:(8)$$q\cdot \nabla \theta =J^{-1}q_{R}\cdot \nabla \;R\;\theta .$$

In connection with the magnetic field H and the magnetization M, we can consider the fields
$$\;H=J^{-1}F^{T}H,\;\;M=JF^{-1}M.$$
The fields $J^{-1}F^{T}H$ and $JF^{-1}M$ are invariant:$${\;H}^{*}={\left(J^{*}\right)}^{-1}{\left(F^{*}\right)}^{T}H^{*}=J^{-1}F^{T}Q^{T}QH=\;H,$$
$${\;M}^{*}=J^{*}{F^{*}}^{-1}M^{*}=JFQ^{T}QM=\;M.$$
Consequently, the scalars
H=|H|,M=|M|,H·M
are also invariant. Indeed, we have
$$H=J^{-1}{\left(F^{T}H\cdot F^{T}H\right)}^{1/2}=J^{-1}{\left(H\cdot BH\right)}^{1/2},$$
$$M=J{\left(F^{-1}M\cdot F^{-1}M\right)}^{1/2}=J{\left(M\cdot B^{-1}M\right)}^{1/2},$$
$$\;H\cdot \;M=F^{T}H\cdot F^{-1}M=H\cdot M,$$
where $B=FF^{T}$. Hence, in addition to being Euclidean invariants, the fields H,M make the inner product H·M invariant and H·M=H·M. Likewise, we found that $H=J\;H=F^{T}H$ is invariant too.

It is worth expressing the power $\mu_{0}\rho H\cdot \dot{m}$ in terms of H and M. Let $\rho_{R}=\rho J$ be the mass density in the reference configuration. Since $M=J^{-1}F\;M$,
$$m=\frac{1}{\rho }M=\frac{1}{\rho_{R}}F\;M$$
whence
$$\dot{m}=\frac{1}{\rho_{R}}\left(\dot{F}\;M+F\dot{\;M}\right)=\frac{1}{\rho_{R}}\left(LF\;M+F\dot{\;M}\right)=\frac{1}{\rho }LM+\frac{1}{\rho_{R}}F\dot{\;M}.$$

It then follows that
$$\mu_{0}\rho H\cdot \dot{m}=\mu_{0}\left(H⊗M\right)\cdot L+\mu_{0}J^{-1}H\cdot F\dot{\;M}.$$

Hence, we obtain
(9)$$\mu_{0}\rho H\cdot \dot{m}=\mu_{0}\left(F^{-1}H⊗F^{-1}M\right)\cdot \dot{E}+\mu_{0}\left(H⊗M\right)\cdot W+\mu_{0}\;H\cdot \dot{\;M}.$$

Incidentally,
(10)$$F^{-1}H⊗F^{-1}M=JF^{-1}F^{-T}\;H⊗F^{-1}M=C^{-1}\;H⊗\;M.$$

For later convenience we notice that, by (Equation 9) and (Equation 10),
(11)$$\mu_{0}\rho_{R}H\cdot \dot{m}=\mu_{0}\left(C^{-1}H⊗\;M\right)\cdot \dot{E}+\mu_{0}J\left(H⊗M\right)\cdot W+\mu_{0}H\cdot \dot{\;M},$$
while
$$JT\cdot L=T_{RR}\cdot \dot{E}+JT\cdot W.$$

## 4. Consistency with the Balance of Angular Momentum

While the fields H and M enjoy Euclidean invariance, we now look for specific requirements induced by (Equation 2). We go back to the form (Equation 5) of the Clausius–Duhem inequality and note that
$$-\dot{\psi }+\mu_{0}H\cdot \dot{m}=\left(-\psi +\mu_{0}H\cdot m\right)\dot{}-\mu_{0}m\cdot \dot{H}.$$

Hence, we let
$$\phi =\psi -\mu_{0}H\cdot m$$
and write inequality (Equation 5) in the form
(12)$$-\rho \left(\dot{\phi }+\eta \dot{\theta }\right)-\mu_{0}M\cdot \dot{H}+T\cdot L-\frac{1}{\theta }q\cdot \nabla \theta =\rho \theta \gamma .$$

To fix our ideas, let
θ,F,H,∇θ
be the set of variables for the functions ϕ,η,T,q, and γ. Computation of $\dot{\phi }$ and substitution result in
$$\begin{array}{c} -\rho \left(\partial_{\theta }\phi +\eta \right)\dot{\theta }+\left(T-\rho \partial_{F}\phi ⊗F^{T}\right)\cdot L-\left(\mu_{0}M+\rho \partial_{H}\phi \right)\cdot \dot{H}-\rho \partial_{\nabla \theta }\phi \cdot \dot{\bar{\nabla \theta }}\\ -\frac{1}{\theta }q\cdot \nabla \theta =\rho \theta \gamma \geq 0. \end{array}$$

The arbitrariness of $\dot{\bar{\nabla \theta }},\dot{\theta }$ and $L,\dot{H}$ implies
$$\partial_{\nabla \theta }\phi =0,\;\eta =-\partial_{\theta }\phi$$
and
$$T=\rho \partial_{F}\phi ⊗F^{T},\;\mu_{0}M=-\rho \partial_{H}\phi .$$

The constraint (Equation 2) results in
(13)$$\mathrm{skw}\;\partial_{F}\phi ⊗F^{T}=\mathrm{skw}\;H⊗\partial_{H}\phi$$
and the requirement (Equation 13) holds if $\partial_{F}\phi$ is related to $\partial_{H}\phi$.

Any field $\tilde{H}$ of the form f(J)H is objective. Hence, we let ϕ depend on F through $E=(F^{T}F-1)/2$ and jointly on F and H through $\tilde{H}$, ${\tilde{H}}_{K}=f\left(J\right)F_{iK}H_{i}$. If $\phi =\phi (E,\tilde{H})$ then
$$\partial_{F}\phi ⊗F^{T}=F\partial_{E}\phi F^{T}+\partial_{{\tilde{H}}_{P}}\phi \partial_{F}{\tilde{H}}_{P}\;F^{T}.$$

Since
$$\partial_{F_{iK}}J=JF_{iK}^{-1},\;\partial_{F_{iK}}{\tilde{H}}_{P}=f^{\prime }H_{P}JF_{iK}^{-1}+fH_{i}\delta_{KP}$$
and
$$\partial_{F_{iK}}E_{PQ}=\frac{1}{2}\left(F_{iQ}\delta_{PK}+F_{iP}\delta_{QK}\right),$$
so
(14)$${\left(\partial_{F}\phi ⊗F^{T}\right)}_{ij}=F_{iP}\partial_{E_{P}Q}\phi \;F_{jQ}+\partial_{{\tilde{H}}_{P}}\phi f^{\prime }H_{P}J\delta_{ij}+f\partial_{{\tilde{H}}_{P}}\phi H_{i}F_{jP},$$
(15)$${\left(H⊗\partial_{H}\phi \right)}_{ij}=f\partial_{{\tilde{H}}_{P}}\phi \;H_{i}F_{jP},$$
where $f^{\prime }=df/dJ$. Notice that
$$F\partial_{E}\phi F^{T}+f^{\prime }JH\cdot \partial_{\tilde{H}}\phi \;1\in \mathrm{Sym}.$$

Consequently, by (Equation 14) and (Equation 15), it follows that the requirement (Equation 13) holds identically for any magnetic field
$$\tilde{H}=f\left(J\right)F^{T}H.$$

Owing to the form (Equation 11) of the power, the pair M,H seems more convenient to describe the magnetic behavior in deformable bodies. That is why we then proceed with the choice of H, i.e., f=1, for the referential magnetic field.

## 5. Thermodynamic Restrictions

The Euclidean invariance suggests that we investigate the Clausius–Duhem inequality (Equation 5) in the Lagrangian description. Hence, we consider *J* times inequality (Equation 5) and use the representations (Equation 7)–(Equation 9) of the powers T·L, q·∇θ, and $\mu_{0}\rho H\cdot \dot{m}$ to obtain
(16)$$\begin{array}{c} -\rho_{R}\left(\dot{\psi }+\eta \dot{\theta }\right)+\mu_{0}H\cdot \dot{\;M}+\left(T_{RR}+\mu_{0}\;C^{-1}H⊗\;M\right)\cdot \dot{E}+J\left(T+\mu_{0}H⊗M\right)\cdot W\\ -\frac{1}{\theta }q_{R}\cdot \nabla \;R\;\theta =\rho_{R}\theta \gamma \geq 0. \end{array}$$

Hereafter, we use the referential fields $\eta_{R}=\rho_{R}\eta$, $\psi_{R}=\rho_{R}\psi$. For later developments, it is convenient to consider the free energy
$$\phi_{R}=\psi_{R}-\mu_{0}H\cdot \;M.$$

Moreover, to save writing, we let
(17)$$T_{RR}:=T_{RR}+\mu_{0}\;C^{-1}H⊗\;M.$$

By (Equation 10) and the definition of $T_{RR}$, we have
$$T_{RR}=J\{F^{-1}TF^{-T}+\mu_{0}\left(F^{-1}H\right)⊗\left(MF^{-T}\right\}=JF^{-1}\left\{T+\mu_{0}H⊗M\right\}F^{-T}.$$

Consequently,
(18)$$T_{RR}\in \mathrm{Sym}\;⟺\;T+\mu_{0}H⊗M\in \mathrm{Sym}.$$

Equation (Equation 16) is then rewritten to read
(19)$$-\left({\dot{\phi }}_{R}+\eta_{R}\dot{\theta }\right)-\mu_{0}\;M\cdot \dot{H}+T_{RR}\cdot \dot{E}+J\left(T+\mu_{0}H⊗M\right)\cdot W-\frac{1}{\theta }q_{R}\cdot \nabla \;R\;\theta =\rho_{R}\theta \gamma \geq 0.$$

The purpose of modeling ferromagnetic hysteresis suggests that we take (θ,F,H,M, $\nabla \theta ,\dot{F},\dot{H})$ as the set of independent variables, or alternatively $\dot{M}$ in place of $\dot{H}$. Yet invariance requirements demand that the dependence on the derivatives occurs in an objective way. Moreover, the Euclidean invariance of the free energy ϕ implies that the dependence of $\phi_{R}$ is a function of Euclidean invariants. Now, θ,E,H,M are invariants, and hence we let
$$\phi_{R}=\phi_{R}\left(\theta ,E,H,\;M,\nabla \;R\;\theta ,\dot{E},\dot{H}\right)$$
and the like for $\eta_{R}$, $T_{RR},q_{R}$, and γ.

Compute the time derivative of $\phi_{R}$ and substitute in (Equation 19) to obtain
(20)$$\begin{array}{c} -\left(\partial_{\theta }\phi_{R}+\eta_{R}\right)\dot{\theta }+\left(T_{RR}-\partial_{E}\phi_{R}\right)\cdot \dot{E}-\left(\mu_{0}\;M+\partial_{H}\phi_{R}\right)\cdot \dot{H}-\partial_{\;M}\phi_{R}\cdot \dot{\;M}-\partial_{\nabla \;R\;\theta }\phi_{R}\cdot \nabla \;R\;\dot{\theta }\\ -\partial_{\dot{E}}\phi_{R}\cdot \ddot{E}-\partial \dot{H}\phi_{R}\cdot \ddot{H}+J\left(T+\mu_{0}H⊗M\right)\cdot W-\frac{1}{\theta }q_{R}\cdot \nabla \;R\;\theta =\rho_{R}\theta \gamma \geq 0.\; \end{array}$$

The (linearity and) arbitrariness of $\nabla \;R\;\dot{\theta },\ddot{E},\ddot{H},\dot{\theta },W$ implies that
$$\partial_{\nabla R\theta }\phi_{R}=0,\;\partial_{\dot{E}}\phi_{R}=0,\;\partial_{\dot{H}}\phi_{R}=0,$$
(21)$$\eta_{R}=-\partial_{\theta }\phi_{R},\;T+\mu_{0}H⊗M\in \mathrm{Sym}.$$

The symmetry condition in (Equation 21) is just the balance relation (Equation 2) of angular momentum. Hence, (Equation 20) simplifies to
(22)$$\left(T_{RR}-\partial_{E}\phi_{R}\right)\cdot \dot{E}-\left(\mu_{0}\;M+\partial_{H}\phi_{R}\right)\cdot \dot{H}-\partial_{\;M}\phi_{R}\cdot \dot{\;M}-\frac{1}{\theta }q_{R}\cdot \nabla \;R\;\theta =\rho_{R}\theta \gamma \geq 0.$$

In the following analysis of (Equation 22), we neglect cross-coupling effects. Specifically, we assume T is independent of $\dot{H}$ and ∇Rθ; $\dot{\;M}$ is independent of $\dot{E}$ and ∇Rθ; $q_{R}$ is independent of $\dot{E}$ and $\dot{H}$. Consequently, inequality (Equation 22) splits into three sub-inequalities: (23)$$-\left(\mu_{0}\;M+\partial_{H}\phi_{R}\right)\cdot \dot{H}-\partial_{\;M}\phi_{R}\cdot \dot{\;M}=\rho_{R}\theta \gamma_{H}\geq 0,$$
(24)$$\left(T_{RR}-\partial_{E}\phi_{R}\right)\cdot \dot{E}=\rho_{R}\theta \gamma_{T}\geq 0,$$
(25)$$-\frac{1}{\theta }q_{R}\cdot \nabla \;R\;\theta =\rho_{R}\theta \gamma_{q}\geq 0.$$

The three functions $\gamma_{H},\gamma_{T},$ and $\gamma_{q}$ are non-negative as particular cases of γ; i.e., $\gamma_{H}$ is the value of γ as $\dot{E}=0,\nabla \;R\;\theta =0$ and the like for $\gamma_{T}$ and $\gamma_{q}$. Equation (Equation 23) is investigated in the next sections; the joint occurrence of $\dot{H}$ and $\dot{\;M}$ result in hysteretic properties of the material. As for Equation (Equation 24), the stress $T_{RR}$, and hence $T_{RR}$, can depend on $\dot{E}$. This dependence is allowed in the form
(26)$$T_{RR}=\partial_{E}\phi_{R}+\Xi \;\dot{E},\;\rho_{R}\theta \gamma_{T}=\dot{E}\cdot \Xi \dot{E},$$
where Ξ is a positive semi-definite fourth-order tensor such that Sym→Sym. In view of (Equation 17), we have
(27)$$T+\mu_{0}H⊗M=J^{-1}F\partial_{E}\phi_{R}F^{T}+\hat{\Xi }\;\dot{E};$$
in suffix notation ${\hat{\Xi }}_{ijRS}=J^{-1}F_{iP}F_{jQ}\Xi_{PQRS}$. Hence, as must be the case, $T+\mu_{0}H⊗M\in \mathrm{Sym}$. Equation (Equation 27) shows that the stress T consists of the elastic term $J^{-1}F\partial_{E}\phi_{R}F^{T}$, the magnetic term $-\mu_{0}H⊗M$, and the viscous term $\hat{\Xi }\dot{E}$. Equation (Equation 25) is the heat equation in the reference configuration. Fourier’s law q=−κ(θ)∇θ is allowed so that
$$q_{R}=-\kappa \left(\theta \right)JC^{-1}\nabla \;R\;\theta$$
and hence makes $\rho_{R}\theta \gamma_{q}=\kappa \left(\theta \right)J\nabla \;R\;\theta \cdot C^{-1}\nabla \;R\;\theta$. Rate-type constitutive equations for $q_{R}$ are obtained by letting ${\dot{q}}_{R}$ be given by a constitutive function while $q_{R}$ is one of the independent variables [19].

### Cyclic Processes

We first go back to inequality (Equation 19) and investigate cyclic processes of inviscid materials, Ξ=0, in uniform temperature fields; ∇Rθ=0. In a cyclic process in the time interval $\left[t_{i},t_{f}\right]$, we have
$$\Gamma \left(t_{i}\right)=\Gamma \left(t_{f}\right),\;\Gamma :=\left(\theta ,E,H,\;M\right).$$

Integration in time of (Equation 19) on $\left[t_{i},t_{f}\right]$ yields
$$\int_{t_{i}}^{t_{f}}\left(-\eta_{R}\dot{\theta }-\mu_{0}\;M\cdot \dot{H}+T_{RR}\cdot \dot{E}\right)dt=\int_{t_{i}}^{t_{f}}\rho_{R}\theta \gamma \;dt\geq 0.$$

Two interesting cases occur in isothermal processes, where $\dot{\theta }\equiv 0$, so that
$$\int_{t_{i}}^{t_{f}}\left(-\mu_{0}\;M\cdot \dot{H}+T_{RR}\cdot \dot{E}\right)dt\geq 0,$$
depending on the constitutive properties. First, if $T_{RR}$ is independent of $\dot{H}$ then both terms are required to be non-negative, so that
(28)$$\int_{t_{i}}^{t_{f}}\left(\;M\cdot \dot{H}\right)dt\leq 0,\;\int_{t_{i}}^{t_{f}}\left(T_{RR}\cdot \dot{E}\right)dt\geq 0.$$

Second, let
$$\phi_{R}=\Phi_{R}\left(\theta ,E\right)+\varphi_{R}\left(\theta ,H,\;M\right).$$

Since
$$-\left({\dot{\phi }}_{R}+\eta_{R}\dot{\theta }\right)-\mu_{0}\;M\cdot \dot{H}+T_{RR}\cdot \dot{E}=-{\dot{\varphi }}_{R}-\left(\partial_{\theta }\Phi_{R}+\eta_{R}\right)\dot{\theta }-\mu_{0}\;M\cdot \dot{H}+\left(T_{RR}-\partial_{E}\phi_{R}\right)\cdot \dot{E}$$
and $T_{RR}-\partial_{E}\phi_{R}=0$, throughout an isothermal cyclic process, we have
(29)$$\int_{t_{i}}^{t_{f}}\left(\;M\cdot \dot{H}\right)dt\leq 0.$$

Of course if $T_{RR}$ satisfies (Equation 26), then the corresponding integral (Equation 28) is non-negative.

## 6. Hyper-Magnetoelastic Materials

If M is not among the independent variables, then the arbitrariness of $\dot{H}$ in (Equation 23) implies
(30)$$\mu_{0}\;M=-\partial_{H}\phi_{R},$$
in addition to $\gamma_{H}=0$. The dependence of $\phi_{R}$ on θ,E,H allows us to say that Equation (Equation 30) represents the constitutive equation of the magnetization in a hyper-magnetoelastic material.

### 6.1. Linear Magnetoelastic Materials

For definiteness, we look for constitutive equations associated with a special class of free energies. Let χ possibly depend on θ. Let be the magnetic susceptibility, per unit volume, in the current configuration. We assume that the free energy ρϕ is the sum of a thermoelastic part ρΨ(θ,C) and a quadratic isotropic part due to magnetization:(31)$$\rho \phi \left(\theta ,C,H\right)=\rho \Psi \left(\theta ,C\right)-\frac{1}{2}\mu_{0}\chi \left(\theta \right){\left|H\right|}^{2}.$$

Replacing $H=F^{-T}H$ and multiplying by J, we find
(32)$$\phi_{R}\left(\theta ,C,H\right)=\Psi_{R}\left(\theta ,C\right)-\frac{1}{2}\mu_{0}\chi J\left(C^{-1}H\right)\cdot H;$$
the form (Equation 32) shows that $\phi_{R}$ is a function of invariant quantities. By (Equation 30), we have
(33)$$\mu_{0}\;M=\mu_{0}\chi JC^{-1}H.$$

Hence, it follows that
$$M=\chi H,\;H={\left(\chi J\right)}^{-1}C\;M,$$
which represents the magnetization function in a linear paramagnetic material. The associated free energy in terms of M is obtained by a Legendre transformation of (Equation 32),
(34)$$\psi_{R}\left(\theta ,C,\;M\right):=\phi_{R}+\mu_{0}H\cdot \;M=\Psi_{R}\left(\theta ,C\right)+\frac{1}{2}\mu_{0}{\left(\chi J\right)}^{-1}\left(C\;M\right)\cdot \;M.$$

Correspondingly, in the current configuration the free energy is
$$\rho \psi \left(\theta ,C,M\right):=\rho \phi +\mu_{0}H\cdot M=\rho \Psi \left(\theta ,C\right)+\frac{1}{2}\mu_{0}\chi^{-1}{\left|M\right|}^{2}.$$

For later convenience, we show that ϕ and ψ can be given by a joint dependence on H and M. Owing to (Equation 33), we can write (Equation 34) as
(35)$$\psi_{R}\left(\theta ,C,H\right)=\Psi_{R}\left(\theta ,C\right)+\frac{1}{2}\mu_{0}\chi J\left(C^{-1}H\right)\cdot H,$$
and then
$$\phi_{R}\left(\theta ,C,H,\;M\right):=\psi_{R}-\mu_{0}H\cdot \;M=\Psi_{R}\left(\theta ,C\right)+\frac{1}{2}\mu_{0}\chi \left(\theta \right)J\left(C^{-1}H\right)\cdot H-\mu_{0}H\cdot \;M.$$

Correspondingly,
(36)$$\rho \phi \left(\theta ,C,H,M\right)=\rho \Psi \left(\theta ,C\right)+\frac{1}{2}\mu_{0}\chi \left(\theta \right){\left|H\right|}^{2}-\mu_{0}H\cdot M.$$

### 6.2. Nonlinear Magnetoelastic Materials

According to Landau’s pioneering approach [20], nonlinear isotropic paramagnets are associated with a free-energy function with a fourth-degree polynomial in the form
$$\rho \psi \left(\theta ,C,H,M\right)=\rho \Psi \left(\theta ,C\right)+\frac{1}{2}\mu_{0}\chi^{-1}{\left(\theta \right)|M|}^{2}+\frac{1}{4}\mu_{0}\kappa {\left|M\right|}^{4},$$
where χ is given by the Curie–Weiss law
$$\chi =\frac{C}{\theta -\theta_{C}},\;C>0,$$
and κ is a positive parameter. Hence, multiplication by *J* results in
$$\psi_{R}\left(\theta ,C,\;M\right)=\Psi_{R}\left(\theta ,C\right)+\frac{1}{2}\mu_{0}{\left(\chi J\right)}^{-1}\left(C\;M\right)\cdot \;M+\frac{1}{4}\mu_{0}\kappa J^{-3}{\left[\left(C\;M\right)\cdot \;M\right]}^{2}.$$

With this free energy, we obtain
$$H={\left(\chi J\right)}^{-1}\left[1+\chi \kappa J^{-2}\left(C\;M\right)\cdot \;M\right]C\;M$$
and
(37)$$H=\left[\frac{1}{C}\left(\theta -\theta_{C}\right)+\kappa {\left|M\right|}^{2}\right]M.$$

When the applied external field H vanishes, it follows that M=0 is a solution at any temperature. In addition, if $\theta <\theta_{C}$, there exist infinitely many pairs of non-vanishing solutions:$$M=\pm M_{s}\left(\theta \right)e,\;M_{s}\left(\theta \right):=\sqrt{\frac{\theta_{C}-\theta }{C\kappa }},$$
where e is a generic unit vector. Hence, $M_{s}\left(\theta \right)$ can be viewed as the spontaneous magnetization modulus at $\theta <\theta_{C}$. In the (θ,M) plane, the curve consisting of the two branches $M=\pm M_{s}\left(\theta \right)$ describes a super-critical pitchfork bifurcation which is typical of second-order phase transitions. In addition, by letting $M_{*}=M_{s}\left(0\right)=\sqrt{\theta_{C}/C\kappa },$ we infer that $M_{s}$ is a decreasing function on $\left(0,\theta_{C}\right]$ so that $0\leq M_{s}\left(\theta \right)<M_{*}$ (see the dashed line in Figure 1).

When the applied external field H does not vanish, we assume that M and H have a common direction and consider the pertinent components, *M* and *H*. Then, (Equation 37) becomes unidimensional in character and gives
$$H=\left[\frac{1}{C}\left(\theta -\theta_{C}\right)+\kappa M^{2}\right]M.$$

The corresponding curve in the (θ,M)-plane is drawn in Figure 1 (solid line) for a given positive value of *H*. For all $\theta \gg \theta_{C}$, there exists only one solution, say $M_{0}\left(\theta \right)$, which approaches zero, whereas for $\theta \ll \theta_{C}$ there are three solutions very close to solutions $0,\pm M_{s}$ of the homogeneous case.

Since the differential magnetic susceptibility, $\chi_{d}$, can be computed as the derivative with respect to *H* of the constitutive function for *M*, we infer
$$\chi_{d}\left(M,\theta \right):=\partial_{H}M={\left(\partial_{M}H\right)}^{-1}=\frac{C}{3C\kappa M^{2}+\theta -\theta_{C}}.$$

Hence, if $\theta \gg \theta_{C}$, then $M=M_{0}\left(\theta \right)$ is negligible and we get the Curie–Weiss law:$$\chi_{d}\left(\theta \right):=\chi_{d}\left(M_{0}\left(\theta \right),\theta \right)\approx \frac{C}{\theta -\theta_{C}}.$$

Otherwise, when $0<\theta \ll \theta_{C}$ we have $M\approx M_{s}^{2}\left(\theta \right)=\left(\theta_{C}-\theta \right)/C\kappa$ so that
$$\chi_{d}\left(\theta \right)\approx \chi_{d}\left(M_{s}\left(\theta \right),\theta \right)=\frac{C}{2(\theta_{C}-\theta )}.$$

Summarizing, the plot of $\chi_{d}\left(\theta \right)$ is given in Figure 2.

## 7. Hypo-Magnetoelastic Materials

We now look at more general non-dissipative models consistent with (Equation 23). We let $\gamma_{H}=0$ but allow M to be an independent variable, whence it follows from (Equation 23) that
(38)$$\partial_{\;M}\phi_{R}\cdot \dot{\;M}=-\left(\mu_{0}\;M+\partial_{H}\phi_{R}\right)\cdot \dot{H}.$$

Let n be a unit vector, n·n=1. Any vector w can be represented as the sum of the longitudinal part, along n, and the transverse part, (1−n⊗n)w,
w=(w·n)n+(1−n⊗n)w.

If the transverse part is undetermined, then we can write
(39)w=(w·n)n+(1−n⊗n)g.
for any vector g. The representation (Equation 39) is now applied in connection with $\dot{\;M}$.

Assume $\partial_{\;M}\phi_{R}\neq 0$. Hence, we let
$$n=\frac{\partial_{\;M}\phi_{R}}{|\partial_{\;M}\phi_{R}|}$$
and find from (Equation 39) and (Equation 38) that
$$\begin{array}{c} \dot{\;M}=\frac{\dot{\;M}\cdot \partial_{\;M}\phi_{R}}{|\partial_{\;M}\phi_{R}{|}^{2}}\partial_{\;M}\phi_{R}+\left(1-\frac{\partial_{\;M}\phi_{R}⊗\partial_{\;M}\phi_{R}}{|\partial_{\;M}\phi_{R}{|}^{2}}\right)g\;\;\\ =-\frac{\partial_{\;M}\phi_{R}⊗\left(\mu_{0}\;M+\partial_{H}\phi_{R}\right)}{|\partial_{\;M}\phi_{R}{|}^{2}}\dot{H}+\left(1-\frac{\partial_{\;M}\phi_{R}⊗\partial_{\;M}\phi_{R}}{|\partial_{\;M}\phi_{R}{|}^{2}}\right)g. \end{array}$$

If g=0, we find
(40)$$\dot{\;M}=M\dot{H},\;M:=-\frac{\partial_{\;M}\phi_{R}⊗\left(\mu_{0}\;M+\partial_{H}\phi_{R}\right)}{|\partial_{\;M}\phi_{R}{|}^{2}}.$$

The second-order tensor M in (Equation 40) depends non-linearly on the strain E and the temperature θ, beyond M and H. By analogy with hypo-elastic materials ([21], §99), we say that Equation (Equation 40) characterizes hypo-magnetoelastic materials.

Otherwise, if $g=\Gamma \dot{H}$, then
$$\dot{\;M}=-\frac{n⊗(\mu_{0}\;M+\partial_{H}\phi_{R})}{|\partial_{\;M}\phi_{R}|}\dot{H}+\left[1-n⊗n\right]\Gamma \dot{H},$$
whence
(41)$$\dot{\;M}=\left[\Gamma -\frac{n⊗(\mu_{0}\;M+\partial_{H}\phi_{R}+\Gamma^{T}\partial_{\;M}\phi_{R})}{|\partial_{\;M}\phi_{R}|}\right]\dot{H}.$$

A particular case follows by taking Γ such that
(42)$$\mu_{0}\;M+\partial_{H}\phi_{R}+\Gamma^{T}\partial_{\;M}\phi_{R}=0,$$
thereby implying the vanishing of the dyadic term. Indeed, inner multiplication of $\dot{\;M}=\Gamma \dot{H}$ by $\partial_{\;M}\phi_{R}$ and the use of (Equation 41) yield (Equation 38).

### A Simple Hypo-Magnetoelastic Model

A special but significant class of hypo-magnetoelastic models is obtained assuming the free energy ψ is independent of M. In this case $\partial_{\;M}\phi_{R}=-\mu_{0}H$ and
(43)$$M\left(\theta ,C,H\right)=\Gamma +\frac{1}{{\left|H\right|}^{2}}H⊗\left[\mu_{0}\;M+\partial_{H}\phi_{R}-\mu_{0}\Gamma^{T}H\right].$$

In the special case Γ=0, it follows that
$$M\left(\theta ,C,H\right)=\frac{1}{{\left|H\right|}^{2}}H⊗\left(\mu_{0}\;M+\partial_{H}\phi_{R}\right).$$

Otherwise, if Γ≠0, we can choose $\Gamma =\hat{\Gamma }\left(\theta ,C,H\right)$ such that (Equation 42) holds as an identity for any value of θ,C,H. From (Equation 43), it follows $M=\hat{\Gamma }$ and then
(44)$$\dot{\;M}=\hat{\Gamma }\left(\theta ,C,H\right)\dot{H}.$$

For definiteness, we exhibit a simple example assuming a quadratic expression of the free energy: $$\psi_{R}=\Psi_{R}\left(\theta ,C\right)+\frac{1}{2}H\cdot \Upsilon_{H}\left(\theta ,C\right)H,\;\Upsilon_{H}=\Upsilon_{H}^{T}.$$

Since
$$\partial_{H}\phi_{R}=\partial_{H}\left[\psi_{R}-\mu_{0}H\cdot \;M\right]=\Upsilon_{H}H-\mu_{0}\;M,$$
condition (Equation 42) reads $\Upsilon_{H}H=\mu_{0}{\hat{\Gamma }}^{T}H$ and the arbitrariness of H finally implies $\mu_{0}\hat{\Gamma }=\Upsilon_{H}$. The special choice $\Upsilon_{H}=\mu_{0}\chi \left(\theta \right)JC^{-1}$, corresponding to the free energy (Equation 36), gives
$$\dot{\;M}=\chi \left(\theta \right)JC^{-1}\dot{H}.$$

## 8. Ferromagnetic Hysteresis

Starting from the dependence on the set of variables
$$\theta ,E,H,\;M,\dot{H}$$
we have found that $\phi_{R}=\phi_{R}\left(\theta ,E,H,\;M\right)$, $T_{RR}=\partial_{E}\phi_{R}$, and (Equation 23) is required to hold with $\gamma_{H}\geq 0$. In addition, in an isothermal cyclic process, the inequality (Equation 29) has to be true. For definiteness, we now investigate hysteresis properties by letting
$$\rho_{R}\theta \gamma_{H}=\zeta \left|\dot{H}\right|,\;\zeta >0.$$

Hence, M and H are subject to
(45)$$\left(\partial_{H}\phi_{R}+\mu_{0}\;M\right)\cdot \dot{H}+\partial_{\;M}\phi_{R}\cdot \dot{\;M}=-\zeta \left|\dot{H}\right|.$$

To select appropriate free energy functions $\phi_{R}$ we observe that, in the hysteretic regime, M and H are neither independent nor are related in the form M=χH, as they are in the paramagnetic regime. If we assume $\partial_{\;M}\phi_{R}\neq 0$ and let
$$n=\frac{\partial_{\;M}\phi_{R}}{|\partial_{\;M}\phi_{R}|},\;g=\Gamma \dot{H}$$
then the requirement (Equation 45) and the representation formula (Equation 39) yield
$$\dot{\;M}=-\frac{n⊗(\mu_{0}\;M+\partial_{H}\phi_{R})}{|\partial_{\;M}\phi_{R}|}\dot{H}-\frac{\zeta |\dot{H}|}{|\partial_{\;M}\phi_{R}|}n+\left[1-n⊗n\right]\Gamma \dot{H},$$
whence
(46)$$\dot{\;M}=\left[\Gamma -\frac{n⊗(\mu_{0}\;M+\partial_{H}\phi_{R}+\Gamma^{T}\partial_{\;M}\phi_{R})}{|\partial_{\;M}\phi_{R}|}\right]\dot{H}-\frac{\zeta |\dot{H}|}{|\partial_{\;M}\phi_{R}|}n.$$

This relation shows a particular case that follows by taking Γ such that
(47)$$\mu_{0}\;M+\partial_{H}\phi_{R}+\Gamma^{T}\partial_{\;M}\phi_{R}=0,$$
thereby implying the vanishing of the dyadic term. By (Equation 47), it follows that the free energy $\phi_{R}$ depends on H and M with a linear relation between $\partial_{H}\phi_{R}$ and $\partial_{\;M}\phi_{R}$. Hence, we let ϕ depend also on ${\left|M-\chi H\right|}^{2}$, and correspondingly $\phi_{R}$ on M and H. Moreover, this term is required to account for the ferromagnetic regime up to the Curie temperature $\theta_{C}$. With this in mind, we generalize the function (Equation 36) to
$$\rho \phi \left(\theta ,C,H,M\right)=\rho \Psi \left(\theta ,C\right)+\frac{1}{2}\mu_{0}{\chi \left(\theta \right)|H|}^{2}-\mu_{0}H\cdot M+\frac{1}{2}\alpha \left(\theta \right)U\left(\theta_{C}-\theta \right){\left|M-\chi H\right|}^{2}$$
where U is the Heaviside step function and α(θ)>0 describes the possible dependence on temperature. In the material description, we have
(48)$$\begin{array}{c} \phi_{R}\left(\theta ,C,H,\;M\right)=\Psi_{R}\left(\theta ,C\right)+\frac{1}{2}\mu_{0}\chi J\left(C^{-1}H\right)\cdot H-\mu_{0}H\cdot \;M\\ +\frac{1}{2}\alpha \left(\theta \right)U\left(\theta_{C}-\theta \right)\left(J^{-1}C\;M-\chi H\right)\cdot \left(\;M-\chi JC^{-1}H\right). \end{array}$$

For ease of writing, we now understand that $\theta \in (0,\theta_{C})$, and hence $U(\theta_{C}-\theta )$ is omitted. Observe that
$$\partial_{H}\phi_{R}+\mu_{0}\;M=\mu_{0}\chi JC^{-1}H-\alpha \chi \left(\;M-\chi JC^{-1}H\right),$$
$$\partial_{\;M}\phi_{R}=-\mu_{0}H+\alpha \left(J^{-1}C\;M-\chi H\right),$$
and hence
$$\partial_{\;M}\phi_{R}=-{\left(\chi J\right)}^{-1}C\left[\partial_{H}\phi_{R}+\mu_{0}\;M\right].$$

Consequently, the constraint (Equation 47) holds with $\Gamma =\chi JC^{-1}$, and the representation (Equation 46) can be written in the form
(49)$$\dot{\;M}=\chi JC^{-1}\dot{H}-\frac{\zeta |\dot{H}|/\alpha }{|J^{-1}C\;M-\left(\chi +\mu_{0}/\alpha \right){H|}^{2}}\left[J^{-1}C\;M-\left(\chi +\mu_{0}/\alpha \right)H\right].$$

### 8.1. One-Dimensional Models of Hysteresis

Assume the spatial fields H and M are collinear and the body is isotropic, or otherwise H and M are in the direction of easy magnetization (easy axis of the transversely isotropic body). We then let $H=He_{1},M=Me_{1}$ and take $\left(e_{1},e_{2},e_{3}\right)$ be an orthonormal basis. Hence, we represent the deformation gradient in the form
F=diag(1+ξ,1−δ,1−δ).

Thus, we have $J=\left(1+\xi \right){\left(1-\delta \right)}^{2}$ and
$$H=F^{T}H=\mathrm{diag}\left(\left(1+\xi \right)H,0,0\right),\;\;M=JF^{-1}M=\mathrm{diag}\left({\left(1-\delta \right)}^{2}M,0,0\right).$$

This allows us to look at a one-dimensional setting. Furthermore we restrict attention to small deformations, i.e., |ξ|,|δ|≪1, and then we assume H and M are approximately equal to H and M. Consequently, we consider the one-dimensional version of (Equation 29) and (Equation 45) in the form
(50)$$\int_{t_{i}}^{t_{f}}M\dot{H}dt=\oint MdH\leq 0,$$
(51)$$\left(\partial_{H}\phi_{R}+\mu_{0}M\right)\dot{H}+\partial_{M}\phi_{R}\dot{M}=-\zeta \left|\dot{H}\right|.$$

Inequality (Equation 50) implies that the closed curve in the H−M plane, associated with the cyclic process, is run in the counterclockwise sense. In rigid bodies, H=H, M=M, and (Equation 51) holds exactly. Provided that $\partial_{M}\phi_{R}\neq 0$, from (Equation 51) it follows that
$$\dot{M}=-\frac{\partial_{H}\phi_{R}+\mu_{0}M}{\partial_{M}\phi_{R}}\dot{H}-\frac{\zeta }{\partial_{M}\phi_{R}}\left|\dot{H}\right|.$$

Now, we consider the one-dimensional version of (Equation 48):$$\phi_{R}=\Psi_{R}+\frac{1}{2}\mu_{0}\chi H^{2}-\mu_{0}HM+\frac{1}{2}\alpha {\left[M-\chi H\right]}^{2},$$
where α=α(θ)>0. Correspondingly, Equation (Equation 49) simplifies to
$$\dot{M}=\chi \dot{H}-\frac{\zeta }{\alpha [M-M(H)]}\left|\dot{H}\right|.$$
where $M\left(H,\theta \right)=\left(\chi +\mu_{0}/\alpha \right)H$. Except at inversion points (where $\dot{H}=0$), we have
(52)$$\frac{dM}{dH}=\chi -\frac{\zeta }{\alpha [M-M(H)]}\mathrm{sgn}\;\dot{H}.$$

If ζ depends on $\dot{H}$ at most through $\mathrm{sgn}\;\dot{H}$, then Equation (Equation 52) is invariant under the time change $t\to t^{*}=ct$, c>0, and then we say that the equation is rate-independent. As a check of consistency, we consider the limit behavior of non-dissipative materials, as is the case in some magnetic nanoparticles (see [22]). This is made formal by letting $\gamma_{H}=0$ and then ζ=0 so that (Equation 52) reduces to
$$\frac{dM}{dH}=\chi ,$$
which represents the differential susceptibility of a paramagnetic material.

Let
(53)$$\chi_{1}=\chi ,\;\chi_{2}=-\frac{\zeta }{\alpha [M-M(H,\theta )]}$$
so we can write Equation (Equation 52) as a differential equation,
(54)$$\frac{dM}{dH}=\chi_{1}+\chi_{2}\;\mathrm{sgn}\;\dot{H},$$
for the unknown function M(H). The second term $\chi_{2}\;\mathrm{sgn}\;\dot{H}$ describes hysteretic effects in that the slope changes depending on the sign of $\dot{H}$. Since $\chi_{2}$ is proportional to ζ, the vanishing of the entropy production $\gamma_{H}$ results in $\chi_{2}=0$. Hence, $\chi_{2}=0$ is said to represent (the limit case of) hysteretic non-dissipative materials, and $\chi_{1}$ represents the slope of the curve M(H) of a paramagnetic substance; possibly, the slope is not constant and depends on the values of *M* and *H*. When $\chi_{2}\neq 0$, we can view (Equation 54) as the positive, differential, magnetic susceptibility. We then require that
$$\chi_{1}>0,\;\left|\chi_{2}\right|\leq \chi_{1}.$$

Since α,ζ>0, $\chi_{2}$ satisfies
(55)$$\chi_{2}\left\{\begin{array}{cc} >0 & \mathrm{if}\;M<M(H,\theta ),\\ =0 & \mathrm{if}\;M=M(H,\theta ),\\ <0 & \mathrm{if}\;M>M(H,\theta ), \end{array}\right.$$
according to the counterclockwise sense required by ∮MdH≤0.

In summary, the model is characterized by the paramagnetic susceptibility $\chi_{1}=\chi$, the hysteretic function ζ, and possibly the temperature-dependent function α. By definition, $\chi_{1}$ is fully determined by the free energy $\phi_{R}$, whereas $\chi_{2}$ depends also on ζ. Hence, different models are obtained by using the same function $\phi_{R}$. The function $\chi_{2}$ is connected with the entropy production through ζ, and as we will see in a while, governs the hysteretic properties of the material.

It is of interest to consider the case
$$\alpha \left(\theta \right)=\{\begin{array}{cc} \alpha_{0}/\left(\theta_{C}-\theta \right), & \mathrm{if}\;\theta \in (0,\theta_{C}),\\ 0, & \mathrm{otherwise}, \end{array}$$
where $\alpha_{0}>0$ possibly depends on *F*. Since $M\left(H,\theta \right)=(\chi +\left[\theta_{C}-\theta \right]\mu_{0}/\alpha_{0})H$,
$$\lim_{\theta \to \theta_{C}}\chi_{2}=0,\;\lim_{\theta \to \theta_{C}}M\left(H,\theta \right)=\chi H.$$
Hence, regardless of the form of ζ, as $\theta \to \theta_{C}$ the curve $M=M(H,\theta_{C})=\chi H$ is just the magnetization curve of a paramagnetic material.

As shown by (Equation 54), the hysteretic function ζ, together with α and the sign of $\dot{H}$, affects the differential susceptibility dM/dH. Indeed, $dM/dH=\chi_{1}+\chi_{2}\mathrm{sgn}\;\dot{H}$ is the effective slope of the magnetization curve in the *H*-*M* plane, and $dM/dH=\chi_{1}$ simply represents the average value of the possible slopes at a fixed point (H,M) of this plane.

### 8.2. Soft Iron Models

Soft magnetic materials are of interest because they are easily magnetized and demagnetized. They have low permanent magnetization (magnetic remanence) and low intrinsic coercivity, but have a high level of saturation and a high Curie temperature. To this class belong soft iron and isoperms, e.g., Fe–Ni–Cu alloys and Mn–Zn ferrites. A model for soft materials is now established within the previous scheme:$$\dot{M}=-\frac{\partial_{H}\phi_{R}+\mu_{0}M}{\partial_{M}\phi_{R}}\dot{H}-\frac{\zeta }{\partial_{M}\phi_{R}}\left|\dot{H}\right|,$$
by assuming
$$\phi_{R}=\frac{1}{2}\alpha {\left[M-M\left(H\right)+\mu_{0}H/\alpha \right]}^{2}+\Lambda \left(H\right)-\mu_{0}MH,\;\alpha >0,$$
$$\zeta =\zeta_{0}{\left[M-M\left(H\right)\right]}^{2},\;\zeta_{0}>0,$$
where M(H) is a monotone increasing function and $\Lambda^{\prime }\left(H\right)=\mu_{0}H\left[M^{\prime }\left(H\right)-\mu_{0}/\alpha \right]$. Then,
$$\partial_{M}\phi_{R}=\alpha \left[M-M\left(H\right)\right],\;\partial_{H}\phi_{R}+\mu_{0}M=\alpha \left[M-M\left(H\right)\right]\left[\mu_{0}/\alpha -M^{\prime }\left(H\right)\right]$$

Hence, we have
$$\frac{dM}{dH}=M^{\prime }\left(H\right)-\frac{\mu_{0}}{\alpha }-\frac{\zeta_{0}}{\alpha }\left[M-M\left(H\right)\right]\mathrm{sgn}\;\dot{H}.$$

Letting
$$\tau_{h}=\frac{\zeta_{0}}{\alpha },\;f\left(H\right)=M\left(H\right),\;g\left(H\right)=M^{\prime }\left(H\right)-\frac{\mu_{0}}{\alpha }=f^{\prime }\left(H\right)-\frac{\mu_{0}}{\alpha }$$
we can write
(56)$$\frac{dM}{dH}=g\left(H\right)+\tau_{h}\left(f\left(H\right)-M\right)\mathrm{sgn}\;\dot{H}.$$

Equation (Equation 56) is consistent with the second law of thermodynamics for a given function *f* and $g=f^{\prime }-\mu_{0}/\alpha$. It is of interest that the constitutive relation (1.1) of [6] is similar to (Equation 56). By analogy with [6], we first consider a function *g* to be piecewise constant, and correspondingly, *f* is piecewise-linear. For definiteness, let $\mu_{0}=1,\alpha >2$, and
$$f\left(H\right)=\left\{\begin{array}{cc} \frac{1}{2}\left(H-1\right)+1 & \mathrm{if}\;H<-1,\\ H & \mathrm{if}\;-1\leq H\leq 1,\\ \frac{1}{2}\left(H+1\right)-1 & \mathrm{if}\;H>1, \end{array}\right.\;g\left(H\right)=\left\{\begin{array}{cc} 1-1/\alpha & \mathrm{if}\;-1\leq H\leq 1,\\ \frac{1}{2}-1/\alpha & \mathrm{if}\;|H|>1, \end{array}\right.$$

In this case, hysteresis cycles are obtained by solving the system
$$\left\{\begin{array}{cc} \dot{H}=\omega H\cos\omega t & \\ \dot{M}=\left[f^{\prime }\left(H\right)-\mu_{0}/\alpha \right]\dot{H}+\tau_{h}\left[f(H)-M\right]\left|\dot{H}\right|. \end{array}\right.$$

Figure 3 shows cyclic processes with large and small amplitudes, corresponding to α=5 and $\tau_{h}=0.3$.

As $\theta \to \theta_{C}$, the parameters $\mu_{0}/\alpha$ and $\tau_{h}$ tend to vanish so that *g* approaches $f^{\prime }$ and (Equation 56) reduce to
$$\frac{dM}{dH}=f^{\prime }\left(H\right).$$

Hence, M=f(H) can be viewed as the magnetization curve of a paramagnetic material.

Some properties, e.g., counterclokwise orientation, are established in [6] by assuming that $f^{\prime }\geq g$. This condition, which here implies $\mu_{0}/\alpha \geq 0$, entails that the energy expended in a complete traversal of a simple loop is non-negative ([7] Equations (1.6) and (3.18)). However, it is not enough to ensure thermodynamic consistency with the existence of a free energy. A stronger requirement that guarantees this consistency is the existence of a positive constant ϵ>0 such that $f^{\prime }-g>\epsilon$ for all *H* and *M*. In the present model, this property is trivially satisfied as $f^{\prime }-g=\mu_{0}/\alpha >0$. Unfortunately, it implies that $f^{\prime }-g$ cannot vanish, not even at the limit as |H| goes to infinity, and this prevents the model from exhibiting the saturation property.

A model allowing for the saturation property can be obtained as follows. Let $\zeta_{0}>0$ and
$$g\left(H\right)=f^{\prime }\left(H\right)-\mu_{0}/\alpha >0,\;\zeta \left(H,M\right)=\zeta_{0}g\left(H\right){\left[f\left(H\right)-M\right]}^{2}.$$

Hence,
$$\frac{dM}{dH}=g\left(H\right)\left[1+\tau_{h}\left(f\left(H\right)-M\right)\;\mathrm{sgn}\;\dot{H}\right].$$

The vanishing of χ=g(H) as |H| approaches infinity is a way of modeling the saturation property. On the other hand, this entails that $f^{\prime }$ approaches $\mu_{0}/\alpha$ as |H| tends to infinity. Hysteresis paths are obtained by solving the system
$$\left\{\begin{array}{cc} \dot{H}=\omega H\cos\omega t & \\ \dot{M}=g\left(H\right)\left\{\dot{H}+\tau_{h}\left[f(H)-M\right]\left|\dot{H}\right|\right\}, \end{array}\right.$$
starting from $\left(H_{0},M_{0}\right)$ with different initial values. In Figure 4, hysteresis cycles are depicted with different amplitudes A=0.4,1.4 and different initial values $H_{0}=0$; $M_{0}=-0.1,\;0,\;0.1$.

As θ→θC, the parameters μ0/α and τh tend to vanish so that *g* approaches f′ and (Equation 56) reduce to
dMdH=f′(H).

Since we assume that g(H) vanishes as |H| approaches infinity, the same does $f^{\prime }\left(H\right)$ in the limit $\theta \to \theta_{C}$. Consequently, M=f(H) can be viewed as the magnetization curve of a paramagnetic material with the saturation property.

### 8.3. Hysteresis Loss

Some remarks are in order on the dissipation due to hysteresis in the general scheme (Equation 51). Owing to the counterclockwise sense,
$$A=-\int_{t_{i}}^{t_{f}}M\dot{H}dt$$
is the area enclosed in a cycle and also $1/\mu_{0}$ times the dissipation of the sample (also called hysteresis loss). For a closed curve in the *H*-*M* plane, it follows from (Equation 51) that
$$A=\frac{1}{\mu_{0}}\int_{t_{i}}^{t_{f}}\zeta \left|\dot{H}\right|dt.$$

If ζ is parameterized by temperature and strain but independent of *H* and *M*, then we can regard ζ as constant in a *H*-*M* cycle so that
$$A=\frac{\zeta }{\mu_{0}}\int_{t_{i}}^{t_{f}}\left|\dot{H}\right|dt=2\zeta \Delta H/\mu_{0}.$$

This is the case for the model (Equation 52), where the dissipation is proportional to the variation ΔH of the magnetic field and twice the hysteretic function ζ.

Otherwise, if ζ is given as in the soft iron model (Equation 56), then
$$A=\frac{\zeta_{0}}{\mu_{0}}\int_{t_{i}}^{t_{f}}{\left[M-f\left(H\right)\right]}^{2}\left|\dot{H}\right|dt,$$
where f=M. Owing to the explicit calculation of the loading and unloading curves that make up the cycle, in ([7], Equation (3.14)) the following result is proved
$$A\left(\Delta H\right)=\frac{2}{\mu_{0}\tau_{h}}\int_{-\Delta H/2}^{\Delta H/2}\left[1-\frac{\cosh(\tau_{h}y)}{\cosh(\tau_{h}\Delta H/2)}\right]\left[f^{\prime }\left(y\right)-g\left(y\right)\right]dy,\;\tau_{h}=\frac{\zeta_{0}}{\alpha };$$
here, for simplicity, we assume $\bar{H}=0$ for the center of the loop. Accordingly, the area of a loop of small amplitude ΔH is of order ${\left(\Delta H\right)}^{3}$,
$$A≃\frac{4\tau_{h}}{3\mu_{0}}\left[f^{\prime }\left(0\right)-g\left(0\right)\right]{\left(\Delta H\right)}^{3},$$
whereas the area of the major loop is given by
$$A_{\infty }=\lim_{\Delta H\to +\infty }A\left(\Delta H\right)=\frac{4}{\mu_{0}\tau_{h}}\int_{0}^{\infty }\left[f^{\prime }\left(y\right)-g\left(y\right)\right]dy.$$

Since all models considered are rate-independent, the hysteresis loss is independent of the frequency at which the alternating magnetic field varies.

### 8.4. A Rate-Dependent Generalization

In order to jointly investigate hysteresis and frequency-dependent dissipation properties, we let
$$\rho_{R}\theta \gamma_{H}=(\zeta_{0}|\dot{H}\left|+\zeta_{1}\right){\left[M-M\left(H,\theta \right)\right]}^{2},\;\zeta_{0},\zeta_{1}>0,$$
where $M\left(H,\theta \right)=\left(\chi +\mu_{0}/\alpha \right)H$ and χ and α possibly depend on θ. Hence, (Equation 51) becomes
$$\left(\partial_{H}\phi_{R}+\mu_{0}M\right)\dot{H}+\partial_{M}\phi_{R}\dot{M}=-(\zeta_{0}|\dot{H}\left|+\zeta_{1}\right){\left[M-M\left(H,\theta \right)\right]}^{2}.$$
Considering once again the one-dimensional version of (Equation 48),
$$\phi_{R}=\Psi_{R}+\frac{1}{2}\mu_{0}\chi H^{2}-\mu_{0}HM+\frac{1}{2}\alpha {\left[M-\chi H\right]}^{2},$$
we obtain
(57)$$\dot{M}=\chi \dot{H}-\frac{1}{\alpha }(\zeta_{0}|\dot{H}\left|+\zeta_{1}\right)\left[M-M\left(H,\theta \right)\right],$$
which represents a generalization of (Equation 56) with f=M and g=χ.

It is easy to check that this rate-type equation is rate-dependent in that the response to an AC magnetic field (i.e., a magnetic field that varies sinusoidally) depends on its frequency. For definiteness, let H(t)=Hsinωt, ω>0. After introducing $t^{*}=\omega t$, we put
$$H^{*}:=H\left(t^{*}/\omega \right)=H\sin t^{*},\;{\dot{H}}^{*}:=\frac{dH^{*}}{dt^{*}}\left(t^{*}\right)=H\cos t^{*}.$$

Then,
$$\dot{H}=\omega {\dot{H}}^{*},\;\dot{M}\left(t\right)=\omega {\dot{M}}^{*},$$
and assuming all parameters are constant, (Equation 57) becomes
$$\omega {\dot{M}}^{*}=\omega \chi {\dot{H}}^{*}-\frac{1}{\alpha }(\omega \zeta_{0}|{\dot{H}}^{*}\left|+\zeta_{1}\right)\left[M^{*}-M\left(H^{*},\theta \right)\right].$$

In the limit of small frequencies, ω→0, the material behaves as a reversible paramagnet, with M=M(H,θ). Hence, $\chi \left(\theta \right)+\mu_{0}/\alpha \left(\theta \right)$ may be considered as the *static* magnetic susceptibility. Otherwise, in the limit of high frequencies, ω→+∞, the ferroelectric material exhibits a frequency-independent hysteresis described by
$${\dot{M}}^{*}=\chi {\dot{H}}^{*}-\frac{\zeta_{0}}{\alpha }\left[M^{*}-M\left(H^{*},\theta \right)\right]\left|{\dot{H}}^{*}\right|.$$

Let ϵ>0. If $\zeta_{0}$ is small enough to satisfy the inequality
$$\zeta_{0}\ll \chi \alpha /\epsilon ,$$
then in the strip |M−M(H)|≤ϵ of the *H*-*M* plane, the material’s behavior is approximately visco-magnetoelastic and obeys the rate equation
$$\dot{M}=\chi \dot{H}-\frac{\zeta_{1}}{\alpha }\left[M-M\left(H,\theta \right)\right].$$

This relation implies that *M* and *H* are not in phase under AC magnetic processes and then are related in a complex form. In addition, a dependence of $\zeta_{0}$ on the derivative $\dot{H}$ (and not only on its sign) would provide the same effect in the general case.

## 9. Generalization to Materials within Non-Uniform Fields

Within a quantum mechanical description, the interaction between magnetic moments is modeled by exchange integrals of the probabilistic densities ([23], Ch. 15). The classical analogue of the interaction in non-uniform fields may be modeled by allowing dependence of the energy on the gradient of the magnetization or of the magnetic field ([20], § 44).

For definiteness, we look for a model involving ∇RH. To account for a dependence on ∇RH, we consider the Clausius–Duhem inequality in the more general form with a possibly non-zero extra-entropy flux $k_{R}$ [24]. Hence, we express the Clausius–Duhem inequality as
$$\begin{array}{c} -\left({\dot{\phi }}_{R}+\eta_{R}\dot{\theta }\right)-\mu_{0}\;M\cdot \dot{H}+T_{RR}\cdot \dot{E}+J\left(T+\mu_{0}H⊗M\right)\cdot W\\ -\frac{1}{\theta }q_{R}\cdot \nabla \;R\;\theta +\theta \nabla \;R\;\cdot k_{R}=\rho_{R}\theta \gamma \geq 0. \end{array}$$
and let
$$\theta ,E,H,\;M,\nabla \;R\;\theta ,\dot{E},\dot{H},\nabla \;R\;H$$
be the set of variables for the constitutive functions of $\phi_{R},\eta_{R},T_{RR},q_{R},\gamma$. The standard computation of ${\dot{\phi }}_{R}$ and substitution into (Equation 19) result in
$$\begin{array}{c} -\left(\partial_{\theta }\phi_{R}+\eta_{R}\right)\dot{\theta }+\left(T_{RR}-\partial_{E}\phi_{R}\right)\cdot \dot{E}-\left(\mu_{0}\;M+\partial_{H}\phi_{R}\right)\cdot \dot{H}-\partial_{\;M}\phi_{R}\cdot \dot{\;M}\\ -\partial_{\nabla \;R\;\theta }\phi_{R}\cdot \nabla \;R\;\dot{\theta }-\partial_{\dot{E}}\phi_{R}\cdot \ddot{E}-\partial \dot{H}\phi_{R}\cdot \ddot{H}-\partial_{\nabla \;R\;H}\phi_{R}\cdot \nabla \;R\;\dot{H}\\ +J\left(T+\mu_{0}H⊗M\right)\cdot W-\frac{1}{\theta }q_{R}\cdot \nabla \;R\;\theta +\theta \nabla \;R\;\cdot k_{R}=\rho_{R}\theta \gamma \geq 0. \end{array}$$

Notice that $k_{R}$ is possibly dependent on $\dot{H}$, and then $\partial_{\nabla \;R\;H}\phi_{R}\cdot \nabla \;R\;\dot{H}$ is not the unique term in $\nabla \;R\;\dot{H}$. The linearity and arbitrariness of $\nabla \;R\;\dot{\theta },\ddot{H},\ddot{E},\dot{\theta }$ imply that
$$\partial_{\nabla \;R\;\theta }\phi_{R}=0,\;\partial_{\dot{H}}\phi_{R}=0,\;\partial_{\dot{E}}\phi_{R}=0,\;\eta =-\partial_{\theta }\phi_{R}.$$

Moreover, the arbitrariness of W implies
$$T+\mu_{0}H⊗M\in \mathrm{Sym}$$
and hence, by (Equation 18), $T_{RR}\in \mathrm{Sym}$. The remaining inequality, divided by θ, reads
(58)$$\begin{array}{c} \frac{1}{\theta }\left(T_{RR}-\partial_{E}\phi_{R}\right)\cdot \dot{E}-\frac{1}{\theta }\left(\mu_{0}\;M+\partial_{H}\phi_{R}\right)\cdot \dot{H}-\frac{1}{\theta }\partial_{\;M}\phi_{R}\cdot \dot{\;M}-\frac{1}{\theta }\partial_{\nabla \;R\;H}\phi_{R}\cdot \nabla \;R\;\dot{H}\\ -\frac{1}{\theta^{2}}q_{R}\cdot \nabla \;R\;\theta +\nabla \;R\;\cdot k_{R}=\rho_{R}\gamma \geq 0. \end{array}$$

The identity
$$-\frac{1}{\theta }\partial_{\nabla \;R\;H}\phi_{R}\cdot \nabla \;R\;\dot{H}=-\nabla \;R\;\cdot \left(\frac{1}{\theta }\partial_{\nabla \;R\;H}\phi_{R}\dot{H}\right)+\left[\nabla \;R\;\cdot \left(\frac{1}{\theta }\partial_{\nabla \;R\;H}\phi_{R}\right)\right]\cdot \dot{H}$$
allows (Equation 58) to be written in the form
$$\begin{array}{c} \frac{1}{\theta }\left(T_{RR}-\partial_{E}\phi_{R}\right)\cdot \dot{E}-\frac{1}{\theta }\left(\mu_{0}\;M+\delta_{H}\phi_{R}\right)\cdot \dot{H}-\partial_{\;M}\phi_{R}\cdot \dot{\;M}-\frac{1}{\theta^{2}}q_{R}\cdot \nabla \;R\;\theta \\ +\nabla \;R\;\cdot \left(k_{R}-\frac{1}{\theta }\partial_{\nabla \;R\;H}\phi_{R}\dot{H}\right)=\rho_{R}\gamma \geq 0. \end{array}$$
where
$$\delta_{H}\phi_{R}=\partial_{H}\phi_{R}-\theta \nabla \;R\;\cdot \left(\frac{1}{\theta }\partial_{\nabla \;R\;H}\phi_{R}\right)$$
is the variational derivative of $\phi_{R}$ with respect to H. Hence, we let
$$k_{R}=\frac{1}{\theta }\partial_{\nabla \;R\;H}\phi_{R}\dot{H}.$$

The remaining inequality is multiplied by θ to read
(59)$$\left(T_{RR}-\partial_{E}\phi_{R}\right)\cdot \dot{E}-\left(\mu_{0}\;M+\delta_{H}\phi_{R}\right)\cdot \dot{H}-\partial_{\;M}\phi_{R}\cdot \dot{\;M}-\frac{1}{\theta }q_{R}\cdot \nabla \;R\;\theta =\rho_{R}\theta \gamma \geq 0.$$

Equation (Equation 59) is strictly analogous to (Equation 23), with the differences being that $\partial_{H}\phi_{R}$ is replaced by $\delta_{H}\phi_{R}$ and the constitutive functions depend also on ∇RH. The analysis of (Equation 23) in Section 5 remains formally unchanged for (Equation 59). We only notice that, by
(60)$$\left(\mu_{0}\;M+\delta_{H}\phi_{R}\right)\cdot \dot{H}+\partial_{\;M}\phi_{R}\cdot \dot{\;M}=-\rho_{R}\theta \gamma_{H},$$
the hysteretic properties are affected by the dependence on ∇RH.

## 10. Relation to Other Models

The literature gives evidence of the Jiles–Atherton model, which in fact has been established in different versions. Here we look at the model described in [12,25].

Denote by $M_{\mathrm{anh}}$ the anhysteretic part of *M* and let
$$M_{\mathrm{anh}}=M_{\mathrm{sat}}L\left(H_{e}/a\right),$$
where $H_{e}=H-\alpha M$ denotes the effective magnetic field, a,α are constants, and L is the Langevin function defined as L(x)=coth(x)−1/x. The magnetization *M* is partitioned into reversible and irreversible parts:(61)$$M=M_{\mathrm{rev}}+M_{\mathrm{irr}}.$$

The connection between $M_{\mathrm{anh}},M_{\mathrm{rev}}$, and $M_{\mathrm{irr}}$ is assumed in the form [12]
(62)$$M_{\mathrm{rev}}=c\left(M_{\mathrm{anh}}-M\right),$$
where *c* is a nonnegative constant also called the domain-wall-bowing parameter.

The irreversible part $M_{\mathrm{irr}}$ is assumed to obey
(63)$$M_{\mathrm{irr}}^{\prime }=\frac{M_{\mathrm{anh}}\left(H\right)-M}{k\;\mathrm{sgn}\;\dot{H}-\alpha \left(M_{\mathrm{anh}}\left(H\right)-M\right)},$$
where a prime ${}^{\prime }$ denotes the derivative with respect to *H* and *k* is a microstructural parameter accounting for pinning and domain-wall motion. In view of (Equation 61)–(Equation 63), we obtain the evolution equation of *M* in the form
(64)$$M^{\prime }=\frac{1}{1+c}\;\frac{M_{\mathrm{anh}}\left(H\right)-M}{k\;\mathrm{sgn}\;\dot{H}-\alpha \left(M_{\mathrm{anh}}\left(H\right)-M\right)}+\frac{c}{1+c}M_{\mathrm{anh}}^{\prime }.$$

Here, the factor c/(1+c) represents the coefficient of reversibility. If k=0, then
$$M^{\prime }=-\frac{1}{\alpha (1+c)}+\frac{c}{1+c}M_{\mathrm{anh}}^{\prime }\left(H\right)=\frac{c}{1+c}\left[M_{\mathrm{anh}}^{\prime }\left(H\right)-\frac{1}{\alpha c}\right],$$
which means that no hysteresis occurs. The right-hand side is a function of *H*, and we let
$$\frac{c}{1+c}\left[M_{\mathrm{anh}}^{\prime }\left(H\right)-\frac{1}{\alpha c}\right]:=\chi_{0}\left(H\right).$$

Hence, the anhysteretic function $M_{\mathrm{anh}}$ is given by
$$M_{\mathrm{anh}}^{\prime }\left(H\right)=\frac{1+c}{c}\chi_{0}\left(H\right)+\frac{1}{\alpha c}.$$

If $\chi_{0}$ is chosen, then $M_{\mathrm{anh}}$ is determined by integration. In particular, as c→∞, we have
$$M^{\prime }\left(H\right)\to M_{\mathrm{anh}}^{\prime }\left(H\right)\to \chi_{0}\left(H\right).$$

Instead, if c=0, then (Equation 64) reduces to
(65)$$M^{\prime }=\frac{M-M_{\mathrm{anh}}\left(H\right)}{\alpha \left(M_{\mathrm{anh}}\left(H\right)-M\right)-k\;\mathrm{sgn}\;\dot{H}}.$$

## 11. Conclusions

Models of ferromagnetic hysteresis are established by following a thermodynamic approach. The class of constitutive properties is required to obey the second law, expressed by the Clausius–Duhem inequality, and the Euclidean invariance. Based on the invariance we have considered $H=F^{T}H$ and $\;M=JF^{-1}M$ as the appropriate magnetic field and magnetization in the constitutive equations. It is worth emphasizing that the selection of material invariant fields is non-unique. The selected pair H,M arises from two features. One is the representation of the standard magnetic power: $$\rho_{R}H\cdot \dot{m}=\left(C^{-1}H⊗\;M\right)\cdot \dot{E}+J\left(H⊗M\right)\cdot W+H\cdot \dot{\;M}.$$

The other one is that the condition
$$T+\mu_{0}H⊗M\in \mathrm{Sym},$$
expressing the balance of angular momentum is assured by the dependence on H,M through H,M. The magnetization field M is a Lagrangian counterpart of M; alternatively, the Lagrangian counterpart of M may be defined as $F^{T}M$ [26].

By applying the representation Formula (Equation 39), we have established the model of hypo- and hyper-magnetoelastic materials.

Ferromagnetic hysteresis is modeled through the thermodynamic condition
$$\left(\partial_{H}\phi_{R}+\mu_{0}\;M\right)\cdot \dot{H}-\partial_{\;M}\phi_{R}\cdot \dot{\;M}=-\zeta \left|\dot{H}\right|,$$
and next with the one-dimensional approximation for small deformations, thereby letting H≃H, M≃M. Moreover, H and M are assumed to be collinear, with H,M being the significant components. The thermodynamic condition
∮MdH≤0
denotes the classical property that the hysteresis curve in the H−M plane is run in the counterclockwise sense. The free energy in the form (Equation 48) has the feature that, through the factor α(θ), the ferromagnetic behavior approaches the paramagnetic one as $\theta \to \theta_{C}$. Hysteresis is shown to be modeled by Equation (Equation 54), where $\chi_{1}$ is the paramagnetic susceptibility and $\chi_{2}$ affects the slope changes depending on the sign of $\dot{H}$. Hence, in general, hysteresis is governed by free energy $\phi_{R}$ and a hysteretic function ζ.

Two definite models have been established for the soft iron. In the first one, the free energy $\phi_{R}$ and the hysteretic function ζ, are quadratic functions; the resulting constitutive equation is similar to Equation (1.1) of [6]. As shown by Figure 3, the saturation does not occur. In the second model, $\phi_{R}$ and ζ are not in polynomial forms, and the saturation property was obtained (see Figure 4).

After discussing the dependence of the hysteresis loss on the quantities involved in an alternating magnetic field, some generalizations of these models were introduced: First, a rate-dependent generalization where hysteresis and frequency-dependent dissipation occur jointly. Second, a generalization to materials within non-uniform fields by allowing a dependence of the energy on the gradient of the magnetization or of the magnetic field.

A future improvement to the theory would be modeling materials where the mechanical hysteresis occurs in connection with the ferromagnetic hysteresis.

## Figures and Tables

**Figure 1 materials-16-02882-f001:**
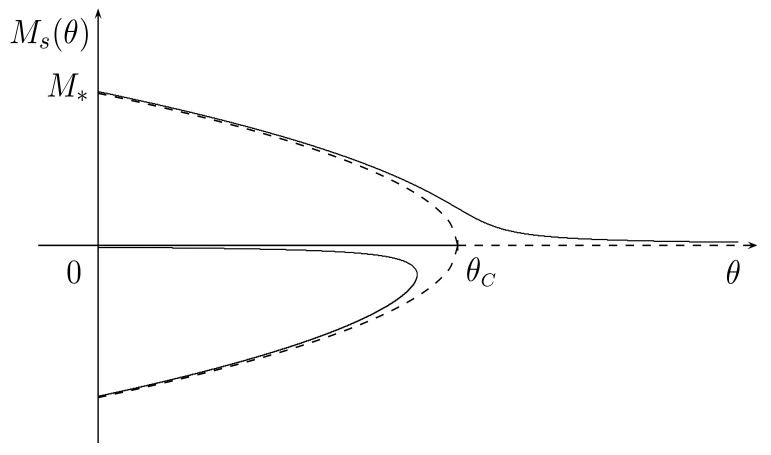
Plot of the perturbed (H=0.05, solid curve) and unperturbed (H=0, dashed curve) super-critical pitchfork bifurcations: C=κ=1.

**Figure 2 materials-16-02882-f002:**
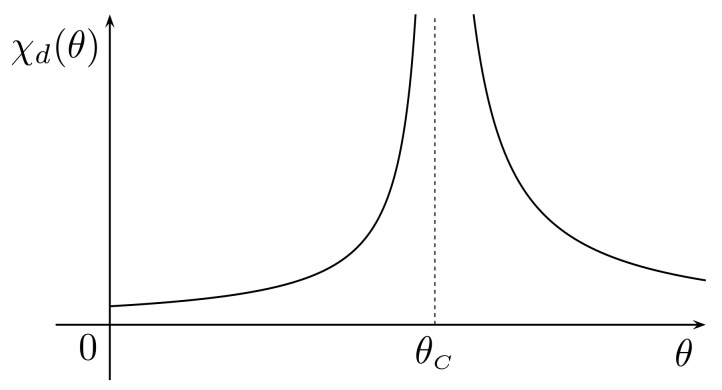
Plot of the (Landau) differential magnetic susceptibility $\chi_{d}$ with C=1.

**Figure 3 materials-16-02882-f003:**
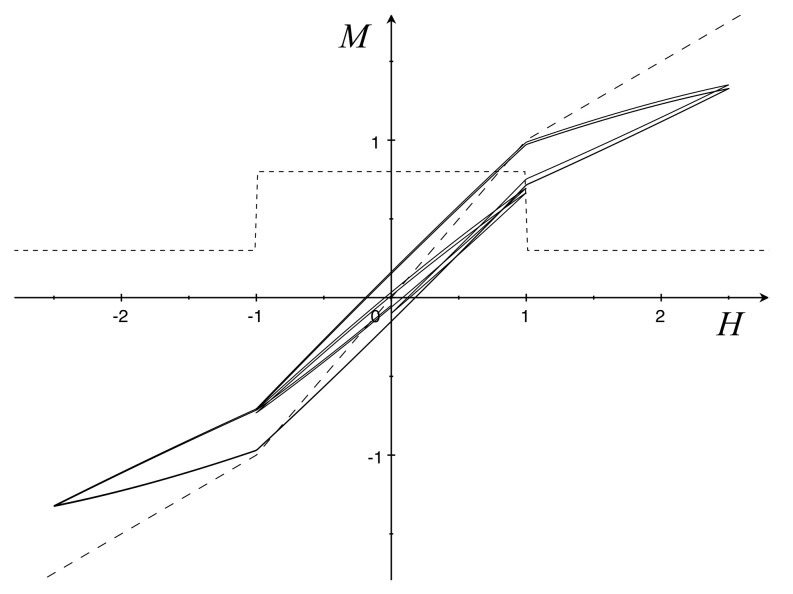
Soft iron hysteresis loops (solid), anhysteretic curve f=M (dashed), and a graph of *g* (short dashed) with H=1,2.5. The initial states are $H_{0}=0$ and $M_{0}=-0.1$.

**Figure 4 materials-16-02882-f004:**
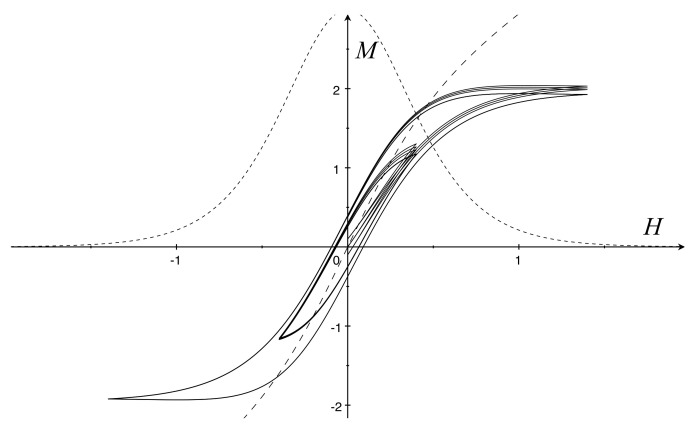
Soft iron hysteresis loops with the saturation properties (solid): $\mu_{0},\tau_{h}=1$, α=2/3, and f(H)=1.5(tanh2H+H) (dashed) and $g\left(H\right)=3/{\left(\cosh2H\right)}^{2}$ (short dashed).

## Data Availability

Not applicable.

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
