# Peer review of "Thermodynamically-Consistent Modeling of Ferromagnetic Hysteresis"

_materials, 2023, doi:10.3390/ma16072882_

Round 1

Reviewer 1 Report

According authors the Duhem-like rate equations seem to be the most convenient scheme to describe any type of hysteresis. Authors of the paper develop a thermodynamic approach to ferromagnetic hysteresis by requiring the consistency of the constitutive functions with the second law expressed by the Clausius-Duhem inequality. Two definite models have been established for the soft iron. In the first one, the free energy and the hysteretic function are quadratic functions. In this case the saturation of magnetization does not occur. In the second one model the free energy and the hysteretic function are not in polynomial forms and the saturation property is obtained.

I do not have any comments or suggestions on the paper.

Reviewer 2 Report

The manuscript titled "A thermodynamically-consistent modeling of ferromagnetic hysteresis" is well written.

The paper are shown hysteretic properties to require that the entropy production is needed in an appropriate form merely to account for a different behavior along the loading or the unloading portion of the loops.

Given all the material, there are several comments:

The monoscript should show which model materials and structures the proposed approach applies to.

1. How does the power absorbed by a sample in an alternating magnetic field depend on the magnetic susceptibility of the sample, the frequency and amplitude of the field?

2. What is the function of the magnetic susceptibility line shape? What does she define?

3. How is the imaginary part of the magnetic susceptibility related to the static susceptibility?

Reviewer 3 Report

The manuscript describes a thermodynamical approach in ferromagnetic materials description. The authors developes thermodynamic theory with the Second law of thermodynamics and Euclidean vector invariance under translations and transformations.

The paper is organized in several sections for step-by-step approach to ferromangetic hysteresis description.

The results shows quite good hysteresis modelling. The further improvment of such analytical models should be significant in the understanding of ferromangetic and mechanical hysteresis materials. Thus the paper is reccomended to be published "as is".

Reviewer 4 Report

In this paper, the author established models of ferromagnetic hysteresis by a thermodynamic approach. The author started from the general thermodynamic relations to allow balance equations and constitutive assumptions involve mechanical and electromagnetic properties. Then, the magnetic field and magnetization are subjected to the standard power representation and angular momentum. The model of hypo- and hyper-magnetoelastic materials and ferromagnetic hysteresis are also established. The manuscript is well written and rigorous. It could be accepted after minor revision on the language and presentation. 

Round 2

Reviewer 2 Report

The authors responded to all comments and corrected the manuscript. I propose to publish the manuscript.